# A Systematic Review of Child Health, Developmental and Educational Outcomes Associated with High Mobility in Indigenous Children from Australia, Canada and New Zealand

**DOI:** 10.3390/ijerph20054332

**Published:** 2023-02-28

**Authors:** Arwen Nikolof, Stephanie J. Brown, Yvonne Clark, Karen Glover, Deirdre Gartland

**Affiliations:** 1Department of Pediatrics, The University of Melbourne, Parkville, VIC 3010, Australia; 2Intergenerational Health, Murdoch Children’s Research Institute, Parkville, VIC 3052, Australia; 3Womens and Kids Theme, South Australian Health and Medical Research Institute, Adelaide, SA 5000, Australia

**Keywords:** residential mobility, moving house, insecure housing, Indigenous, families, children, health, social and emotional, development

## Abstract

Indigenous families tend to move house more often, especially families with young children. However, little is known about the impact of high mobility on children’s well-being and development. The aim of this systematic review was to examine the relationship between residential mobility and children’s health, developmental, and educational outcomes for Australian, Canadian, and New Zealand Indigenous children (0–12 years). Four databases were investigated with pre-determined inclusion and exclusion criteria. The search identified 243 articles after independent screening by two authors. Eight studies assessing four child health outcomes were included, six quantitative and two qualitative. Child health outcomes were classified into four broad categories—physical health, social and emotional behavior, learning and development, and developmental risk. The review identified limited evidence; possible links were identified between high mobility and emotional and behavioral difficulties for younger children. One study identified evidence of a linear relationship between the number of houses a child has lived in since birth and developmental risk. Further research is needed to fully understand the impact of high residential mobility for Indigenous children at different developmental stages. Prioritizing the involvement, collaboration, and empowerment of Indigenous communities and leadership is critical for future research.

## 1. Introduction

A child’s access to safe, stable, and adequate shelter is recognized as a basic human need and is also important for children’s physical health and mental health [1]. Multiple aspects of housing can affect children’s outcomes, including home ownership, housing affordability, mobility (frequency of house moves), homelessness, overcrowding, and poor housing conditions [1]. This review will focus on the consequences of high housing mobility for Indigenous families in Australia, Canada, and New Zealand in terms of the health, developmental, and educational outcomes of children aged 0–12 years.

Moving house in childhood is common. Australia, Canada, and New Zealand comprise some of the most mobile countries in the world, with 40–55% of their population changing where they live every five years [2]. In developing countries, children aged one to four years have the highest rates of mobility. Rates decline towards adolescence and increase again in the early twenties before decreasing again through adulthood. This pattern can expose young children to very high levels of housing mobility [3], with families on low incomes moving more [4]. Low-income families are also more likely to have involuntary or forced moves, for example, where a rental lease is cut short, house payments become unaffordable, or family circumstances change through job loss, family violence, separation, or divorce [4,5,6,7]. There is also evidence that Indigenous people move at higher rates than non-Indigenous populations [4,8,9,10].

The theoretical foundations connecting residential mobility with children’s development are underpinned by an ecological, developmental systems perspective [11]. Developmental contexts in which the child regularly interacts are important and can include family and kin relationships, community, child care, kindergarten, school, and the neighborhood [12]. Children with high mobility can experience multiple changes in these developmental contexts, which in turn, can impact their health and development. With each additional move, families are faced with new family routines, rebuilding social networks, and disruption to children’s education and peer relationships [13]. The culmination of all of these changes impacts families’ and children’s outcomes.

There is a large body of literature showing that high rates of housing mobility have negative consequences for a wide range of child outcomes, including children’s development, cognition, and physical and mental health outcomes [3]. A recent meta-analysis of 844 studies published between 1989 and 2020 reported that the effects of high mobility might be cumulative, with three or more moves in a child’s lifetime more than twice as detrimental to child health as low mobility (one to two moves) [12]. No research was identified examining the potential benefits of mobility for children.

However, there is a wide variation in the way researchers define housing mobility and no agreed definition [14,15]. Key dimensions used to define mobility include the number of house moves over a given period and the age of the child when moves take place [3]. The majority of quantitative studies categorize childhood mobility by the number of house moves over a given period of time [12]. This period can vary from 12 months, 2 years, 5–10 years, or even an 18-year period. The age of children when moving and the age when health outcomes are examined varies between studies. For example, a parent or carer may be asked the number of times their family has moved house between ages 0 and 5 years and then asked about their child’s health outcomes at 6 to 7 years. Child health outcomes can be examined through parent/carer reports or directly through child assessments.

The meta-analysis undertaken by Simsek and colleagues [12] assessed associations between childhood housing mobility and health-related outcomes. In most of the included studies (63 out of 64), childhood mobility was defined as the ‘number of house moves’, however ‘high mobility’ was categorized in different ways; 23 studies measured the ‘number of moves’ on a continuous scale, 26 as a binary high/low mobility and 22 in categories, for example (none, 1–2, 3, or more). In total, 35 studies differentiated between no moves versus one to two moves, and 16 studies between no moves versus three or more moves [12].

Australian research on childhood mobility and its impacts on health and development is limited. Findings for non-Indigenous children suggest early childhood experiences of mobility (0 to 5 years) may have long-term consequences for a range of outcomes, including cognitive development, receptive vocabulary, and emotional and behavioral problems [15,16,17].

Most international research draws on data from majority group populations and non-Indigenous population groups [12,14]. For example, only 6 out of 64 studies from the meta-analysis by Simsek and colleagues focused on ethnic minority populations [12]. Little is known about the health and well-being impacts for Indigenous populations experiencing childhood housing mobility, and there have been numerous calls for greater understanding [12,15,16,18].

Considering housing is a key social determinant of health, increased knowledge in this area will benefit Indigenous communities by driving policy and service change in housing, health, and education. It is important to acknowledge and address Indigenous experiences of mobility to understand the circumstances and lived realities of Indigenous families. If more is known about the health implications of high mobility, strategies may be put in place to better support families and children affected. Such strategies would need to be informed and guided by Indigenous families and involve Indigenous workers across the education, health, and housing sectors.

This paper describes a systematic review of academic and grey literature from Australia, New Zealand, and Canada to examine the relationship between high housing mobility and Indigenous children’s health and development at ages 0–12 years.

The objectives of the review are to (i) identify how housing mobility is defined and measured and (ii) examine the health, developmental, and educational outcomes of Indigenous children aged 0–12 years who experience high housing mobility.

### 1.1. Definitions

#### First Nations Peoples

The review refers to Indigenous groups collectively, as well as describing community and population groups within each study. Australia’s First Nations will respectfully be referred to as Indigenous Australians, encompassing Aboriginal and Torres Strait Islander peoples. This review will also discuss Indigenous Canadians encompassing North American Indian, Métis, and Inuit and Indigenous New Zealand peoples known as Māori.

## 2. Materials and Methods

### 2.1. Search Strategy

A literature search was conducted using key terms associated with housing mobility and health outcomes of Indigenous children according to the inclusion/exclusion criteria. Four databases were searched (ERIC—EBSCO, OVID Medline, Psyc INFO, and PubMed) using search terms developed in collaboration with a research librarian and drawing on other systematic reviews in non-Indigenous populations. Search terms included:

(“Aborigin*” OR “Indigenous”) AND (“Australia*” OR “New-Zealand*” OR “Canada*”) AND “Transient*” OR “migrant*” OR “migration” OR “housing” OR “mobility” OR “residential-instability” OR “geographic-relocation” OR “residential-relocation” OR “residential-stability”.

The search included English language studies published from January 2000 to November 2022. Reference lists of included research articles or related research were examined to identify other relevant papers. The last screening search was conducted in November 2022.

### 2.2. Study Selection

#### 2.2.1. Inclusion Criteria

Articles were included based on the following criteria:Original research including cross-sectional, prospective, and retrospective cohort studies, case-control studies, randomized trials, record linkage studies, and qualitative studies. Grey literature was included once reference lists of included research articles were searched;Indigenous or First Nation children aged 0–12 years from Australia, Canada, and New Zealand were included in the study population;Studies providing a definition of ‘high mobility’ and identifying children and families with high mobility;Studies reporting child health outcomes examined by mobility. Child health outcomes were broadly defined to include development, physical health, mental health and well-being, social and emotional well-being, cognition, behavioral issues, and academic outcomes.

Indigenous children from these three high-income countries were selected as they have important similarities. All three have a system of universal health insurance providing free or subsidized access to primary care and a history of colonization and oppression of their Indigenous peoples, which is reflected in ongoing health disparities between Indigenous and non-Indigenous populations. The review refers to Indigenous groups collectively; however, we acknowledge the vast and rich cultural diversity that exists among and within these populations and communities.

#### 2.2.2. Exclusion Criteria

The following types of studies were excluded: case studies, reviews or commentaries, and studies not including Indigenous children in the age range of 0 to 12 years.

### 2.3. Data Extraction

Papers were imported and actions were recorded in EndNote. Two reviewers (AN, DG) independently screened all titles and abstracts against the inclusion/exclusion criteria. Where there was a difference of opinion, full-text articles were obtained and discussed, and a shared decision was reached.

Data were extracted by a single reviewer using an investigator-developed RED-Cap database and included study characteristics, exposure measures, and child outcomes.

### 2.4. Critical and Cultural Appraisal

Study quality was assessed by two independent reviewers (AN, DG) using the Critical Appraisal Skills Program (CASP) Cohort Study and Qualitative Studies Checklists. Each checklist is designed to measure the validity and applicability of the research and results using a checklist of twelve questions for cohort studies and nine questions for qualitative studies [19]. The cultural competence of the research was assessed by two reviewers (AN, DG) using the Aboriginal and Torres Strait Islander Quality Appraisal Tool [20]. The Aboriginal and Torres Strait Islander QAT consists of fourteen questions that assess the quality of health research from an Indigenous Australian perspective. The questions encompass appropriate research questions; community engagement and consultation; research leadership and governance; community protocols; intellectual and cultural rights; the collection and management of research material; Indigenous research paradigms; a strength-based approach to research; the translation of findings into policy and practice; benefits to participants and community involved; capacity strengthening and two-way learning [21].

## 3. Results

The systematic search identified 321 articles, 8 of which were identified through reference screening. Of these, 78 were duplicates, leaving 243 articles to screen the title and abstracts. The major reasons for excluding papers were: data on mobility not reported; findings for Indigenous children from Australia, Canada, and New Zealand not reported; not examining child health outcomes by mobility. After a full-text review, eight articles were deemed to fit the inclusion criteria (see Figure 1). PROSPERO Registration: CRD42022311367. Results from quantitative and qualitative studies have been reported separately.

### 3.1. Study Characteristics

All studies (quantitative and qualitative) recruited participants from community settings. Three studies recruited children living in urban environments [22,23,24], one recruited children living in remote areas [25], and four included children living in urban, regional, and remote areas [26,27,28,29].

#### 3.1.1. Quantitative Studies

The study characteristics of the six quantitative studies are shown in Table 1. Five were undertaken in Australia and one in Canada. No studies were identified from New Zealand. A total of 13,429 Indigenous children aged between 0 and 17 years took part across the six studies. The sample size across the six studies ranged from 725 to 3993 Indigenous children.

Table 2a describes the definition of high mobility used by each study. Three different definitions of high mobility were used across the six studies. High mobility was defined as: (1) one or more moves per year of child age [28,29]; (2) living in five or more homes over their lifetime [26,27]; (3) living in four or more homes over their lifetime [22,23]. Based on the definition of high mobility being used and the study population, the proportion of Indigenous children experiencing high residential mobility ranged from 10% [29] to 37% [22] across the six studies.

The Guévremont study used data from the Canadian 2006 Aboriginal Children’s Survey (ACS), a national survey of Métis and Inuit families living off-reserve [29]. The survey was designed to provide a picture of the early development of Indigenous children, including social and living conditions. It was developed by Statistics Canada and conducted jointly with Human Resources and Social Development Canada [30]. A sample of children younger than six years was selected from households identified in the 2006 Canadian Census. Information was provided by a parent or guardian in families living in urban, rural, and northern locations across Canada. Children living on First Nation reserves were not included in the survey sample; thus, the results do not apply to the on-reserve population, which accounted for an estimated 43% of Canadian Indigenous people in 2006.

Two studies utilized data from the Western Australian Aboriginal Child Health Survey (WAACHS), a cross-sectional survey undertaken in 2001 and 2002 [26,27]. The survey—funded by the Australian Government—was designed to build knowledge of the social, emotional, academic, and vocational well-being of Aboriginal and Torres Strait Islander children and young people. Dwellings with Aboriginal and/or Torres Strait Islander children or teenagers aged between 0 and 18 years were eligible to take part. A representative population-based sample of 5289 Aboriginal and Torres Strait Islander children, their families, and communities in Western Australia (WA) participated in household interviews. Sample weights were created to reflect the entire WA Aboriginal and Torres Strait Islander population aged under 18 years by weighting each observation by sex, age, and level of relative isolation of residence (a measure of remoteness designed for Aboriginal and Torres Strait Islander people). A total of 10% of children surveyed lived in areas of extreme isolation, 44% lived in areas with some level of isolation, and 34% lived in an urban setting. Although the sample included children older than 12 years, most children included in the study were aged 4–12 years, and the findings were reported by age bracket.

Chando et al. and Williamson et al. both report on baseline cross-sectional data from The Study of Environment on Aboriginal Resilience and Child Health (SEARCH), a longitudinal cohort study of urban Aboriginal children in New South Wales, Australia [22,23]. SEARCH focuses on community-identified health priorities, including healthy development, ear health, social and emotional well-being, and children being placed into out-of-home care and housing. The study was conducted in partnership with four urban and regional Aboriginal Community Controlled Health Services (ACCHS). Children and adolescents aged 0–17 years and their parent/caregivers who attended the services were invited to participate. Phase 1 data collection took place from 2006 to 2012. Williamson et al. focused on caregivers’ reports of their child’s mental health using the standard Australian version of the Strengths and Difficulties Questionnaire (SDQ) [22]. Data on 1005 Aboriginal children aged 4–17 years were examined, including by age groups (4–7 years and 8–11 years) [22]. Chando et al. focused on developmental risk using the Parents’ Evaluation of Development Status (PEDS) completed by 725 caregivers of their child aged 0–8 years.

Data from the Longitudinal Study of Indigenous Children (LSIC) were utilized by Dockery et al., who accessed data on child health outcomes and residential mobility [28]. Two waves of LSIC data collected in 2008 and 2009 from 11 sites across Australia were chosen to mirror the distribution of the Indigenous population across Australia. Most families in the study were recruited using addresses provided by an Australian Government Health Service. Other informal means of contact, such as word of mouth, local knowledge, and study promotion, were also used to supplement the number of children recruited. Families lived in a mix of urban, regional, and remote areas.

#### 3.1.2. Qualitative Studies

The study characteristics of the two qualitative studies are shown in Table 1. Both qualitative studies were undertaken in Australia [24,25]. Although housing mobility or the number of times participants moved was not measured in the studies, moving from house to house was reported as a key theme by Lowell et al. [25], and secondary homelessness a key theme by Anderson et al. [24]. Lowell et al. refer to high mobility as insecure housing and Anderson et al. as precarious housing. Participants in the focus groups use different terms, such as secondary homelessness, frequent relocation, and house-hopping [24].

Lowell et al. explored the strengths and challenges related to early childhood in a remote Northern Territory community in East Arnhem Land, using a culturally responsive qualitative research process with six Yolŋu (Aboriginal) families and their children [25]. Case studies were conducted over five years, combining in-depth interviews with video-reflexive ethnography. Video recordings of participants in their natural environment (ethnography) enabled a ‘reflexive’ process with participants active in exploring video footage. Interviews were also conducted with community members from other clan and family groups, with ages ranging from 18 to 70 years. Community members were invited to participate to ensure diversity in age, clan group, socio-cultural, and educational background. The study was included as insufficient housing emerged as the greatest challenge families experienced in ‘growing up’ their children.

Anderson et al. noted that the majority of Aboriginal Australians live in urban areas, yet most research into housing and health has been conducted in remote communities [24]. They conducted four focus groups to explore the views of Aboriginal people living in urban Western Sydney, Australia. Adult participants reflected on how housing issues, including mobility, affect Aboriginal children. Thirty-five of the thirty-eight participants were Aboriginal. Adult clients and staff were recruited from an Aboriginal community-controlled health service, selected to include particular ages, genders, life stages, health, and socio-economic circumstances. Participants were asked about their housing circumstances and what relationships, if any, they perceive between housing and health.

### 3.2. Defining and Measuring Mobility in the Studies (Aim 1)

#### 3.2.1. Quantitative Studies

All quantitative studies used parent/caregiver reports to measure mobility based on the number of residential moves or the number of houses families had lived in, with high mobility categorized in a number of ways.

Dockery et al. and Guévremont et al. defined mobility based on the number of homes a child had lived in since birth, dividing the child’s age to create a measure of moves per year of age [28,29]. High mobility was defined in both studies as one or more moves per year of child age. Dockery et al. reported that 20% of Australian Indigenous children aged 0–7 years had high mobility (n = 1687 across two waves) [28]. In Canada, Guévremont et al. reported that 10% of Indigenous children aged 2–5 years had high mobility (n = 3640) [29].

The two Zubrick et al. studies [26,27] categorized children living in five or more homes since birth as experiencing high mobility and reported that just over a quarter of Aboriginal children aged 4–17 years had high mobility (27.4%; CI: 25.2–29.6%) [26].

Both Williamson et al. and Chando et al. categorized children living in four or more homes since birth as experiencing high mobility. Williamson et al. reported that 37% of children aged 4–17 years had high mobility [22], and Chando et al. reported that 15% of children aged 0–8 years had high mobility [23]. Chando et al. also reported on the number of houses lived in since birth (1, 2, 3, 4+) as a continuous variable. Results compared children aged 0–8 years who had lived in a single house (24%) with children who had lived in two (21%), three (14%), or four or more houses since birth (15%) [23].

#### 3.2.2. Qualitative Studies

Participants in both studies raised housing mobility as a key theme. In the Anderson et al. study, participants from focus groups reported frequent relocation, having limited housing options, and difficulty accessing housing [24]. Secondary homelessness, the need for transient or emergency accommodation such as living temporarily with family or friends, was a key theme derived from focus groups.

Lowell et al. reported on mobility in terms of housing insecurity and reported that all six participating families moved between houses—some many times during the five-year period of the study [25]. Their options were limited to a single room in someone else’s house or sometimes a tent outside when others had priority for the rooms inside.

### 3.3. Child Health Outcomes Reported in Included Studies (Aim 2)

Child health outcomes were classified into three broad categories—physical health, social and emotional behavior, and learning and development. The outcome measures used in the quantitative studies are detailed in Table 2a. For the qualitative studies, themes raised by the participants have been extracted and organized into the same categories in Table 2b.

### 3.4. Physical Health

#### 3.4.1. Quantitative Studies

Two studies reported on caregiver ratings of their child’s general health using an SF-36 global health item dichotomized into excellent/very good versus good/fair/poor health [28,29]. Guévremont et al. reported that Indigenous children aged 2–5 years who had experienced high mobility (n = 364, 10%) were less likely to be rated by their caregiver as having excellent or very good health compared to children with low mobility. Two multivariable models were reported. The first model adjusted for other housing characteristics (house in need of repair, unaffordable housing, homeownership status, having a regular smoker in the home, and crowding). Children with high mobility had just over half the odds of excellent or very good health (Adj.OR 0.63, *p* = 0.005) compared to children with low mobility. The second model additionally adjusted for socio-demographic variables (parents’ highest level of education, household income, single-parent family, urban residence, and child’s sex and age). High mobility remained significantly associated with lower odds of excellent or very good health (Adj.OR 0.65, *p* = 0.012). Confidence intervals were not reported.

In contrast, using the same measure, Dockery et al. reported families with high mobility (one or more moves per year of age) were 20% less likely to report their 0–7-year-old child as having poor health, but this difference was not statistically significant (β = 0.8, *p* = 0.270). Again, confidence intervals were not reported.

Guévremont et al. was the only study to report on other physical health outcomes associated with high mobility [29]. Caregivers were asked to indicate if their child had experienced a physical health issue in the past year (yes/no), including limited activity, two or more chronic conditions, a serious injury, two or more ear infections, or any chronic respiratory condition (including allergies, asthma, and/or bronchitis). The authors report that children with high mobility were more likely to have a physical activity limitation, two or more chronic conditions, and a chronic respiratory condition compared with children not experiencing high mobility. However, the data underpinning these statements were not reported. More detail was provided regarding ear infections. There was no difference in the prevalence of two or more ear infections for children with high versus low mobility after adjusting for other housing issues (house in need of repair, unaffordable housing, homeownership status, having a regular smoker in the home, and crowding) (Adj OR 1.06, *p* = 0.765). For each of these papers, confidence intervals or standard errors would have provided a greater understanding of the precision of reported results.

#### 3.4.2. Qualitative Studies

Participants in the two qualitative studies identified a range of physical health outcomes. Health outcomes were reported in relation to a range of housing circumstances, including insecure housing, poor housing conditions, and overcrowding [24,25]. Although housing mobility was not the sole focus of these papers, they were included due to the closely connected nature of other housing circumstances with housing mobility [7].

In the Anderson et al. study, participants reported that overcrowding contributed to higher rates of communicable diseases for children, including colds and flu, gastroenteritis, ear, chest, and skin infections, as well as inadequate sleep and food [24]. Otitis media was emphasized as a housing-related illness of concern due to its prevalence and effects on children’s hearing, speech, language, behavior, and education outcomes. Poor housing conditions such as mold and damp were reported to be associated with childhood asthma and respiratory conditions, as well as increased injury risk.

Sharing sickness between household members was one of six key themes reported by Lowell et al. [25]. Parents, family, and community members describe children aged 0–6 years catching scabies, sores, flu, and other infections due to overcrowding. Limited cooking utensils, plates, bedding, and access to a washing machine, bathroom, and kitchen made it difficult for adults to limit the impacts of sharing a crowded house on their children. Adults reported running out of food due to their housing circumstances, which in turn impacted their children’s health.

### 3.5. Social and Emotional Outcomes

#### 3.5.1. Quantitative Studies

Four studies reported on associations between high mobility and children’s social and emotional behavior [22,26,28,29]. Three of the four studies [22,26,27] used the parent/caregiver SDQ, one study reported on individual scales [29], and two studies on the total difficulties score [22,27]. Guévremont et al. used the SDQ for Canadian Indigenous children aged 2–6 years [29]. The authors reported on prosocial behavior, hyperactivity/inattention, emotional symptoms, and conduct problems. After adjusting for housing characteristics (house in need of repair, unaffordable housing, homeownership status, having a regular smoker in the home, and crowding), children with high mobility scored lower on prosocial behavior (β = −0.06, *p* = 0.003) and higher on inattention–hyperactivity (β = 0.19, *p* < 0.001). Within the text, the authors report that children with high mobility scored higher on emotional and conduct problems, but no data were provided. Once parental education was added to the model, the association between mobility and prosocial behavior was no longer significant (β = −0.03, *p* = 0.113), but the association between high mobility and inattention–hyperactivity remained (β = 0.17, *p* = 0.001).

Zubrick et al. used a modified SDQ tailored for Australian Indigenous children (including some changes in item wording and the response scale) [26]. A total difficulties score of ≥17 was used to identify children with a high risk of clinically significant emotional or behavioral difficulties. Children aged 4–17 years with high mobility (5 or more homes since birth) had odds one and a half times higher of clinically significant emotional and/or behavioral difficulties compared to children with low mobility (OR 1.54; 95%CI: 1.07–2.04). This association remained after adjusting for child age and residential isolation (Adj.OR = 1.51; 95%CI: 1.05–2.17). In terms of the mean SDQ score, children with high mobility scored higher (mean = 12.1; 95%CI 11.4–12.8) than children with low mobility (mean = 11.0; 95%CI 10.6–11.4) [26]. Almost one-third of the younger children (aged 4–11) with high mobility had clinically significant difficulties compared to around a quarter of children with low mobility (32.0%, CI: 26.6–37.7% and 24.5%, CI: 22.0–27.3%, respectively).

In Williamson et al., the authors classified children as having ‘good’ mental health if they scored < 17 on the SDQ total difficulties score—the threshold for high risk of emotional or behavioral difficulties [22]. After adjusting for demographic, child, and carer factors, high mobility (>four homes since birth) was significantly associated with lower odds of good mental health for 4–17-year-olds. Analyses were undertaken by three age groups; high mobility in children aged 4–7 years was associated with lower odds of good mental health (OR = 0.39, 95%CI 0.18–0.85). There was no difference in outcomes of children experiencing high residential mobility in the 8–11 and 12–17 year age groups.

In the final study, Dockery et al. used two questions about social and emotional behavior from the Parents Evaluation of Development Status (PEDS): “Do you have any concerns about how (study child) behaves?” and “Do you have any concerns about how (study child) gets along with others?”. Response options were yes/a little/no. After adjusting for child sex and socio-economic status, the authors report that families with high mobility were 32% more likely to report social and emotional behavior concerns for their child, but it was not a statistically significant difference (β = 1.32, *p* = 0.251).

#### 3.5.2. Qualitative Studies

Both qualitative studies reported families experiencing stress due to housing insecurity and overcrowding and the impacts of this stress on their children’s social and/or emotional outcomes [24,25]. The “negative influence of others in the house” was one of six key themes reported by Lowell et al. [25]. Parents, family, and community members highlighted conflict caused by overcrowding and the negative influence of others in the household significantly impacting their children’s behavior. Participants described feeling disempowered and felt that they had “little if any control over the conditions that they know will influence the health and well-being of their children” [25]. Lowell et al. reported that insufficient and insecure housing underpinned conflict among families and between children, causing families to move multiple times [25].

### 3.6. Learning and Development Outcomes

#### 3.6.1. Quantitative Outcomes

School factors were examined in two quantitative studies using different measures [27,28]. Zubrick et al. report on school attendance, academic performance, and high mobility for 2379 Aboriginal and Torres Strait Islander children aged 4–17 years [27]. School attendance was quantified as the median number of days absent per school year (principal report), with children absent from school for 26 days or more (the median) classified as having ‘poor attendance’. Almost half of all Aboriginal students in the cohort had 10 or more unexplained days absent. High mobility was not associated with poor attendance—students who had lived in five or more houses since birth were less likely (OR 0.64; 95%CI: 0.50–0.83) to have had more than 10 days of unexplained absence from school. The majority of students with poor attendance (67.5%) or unexplained absences were classified as having low academic performance, with academic performance declining systematically with any absence from school [27].

Several measures of academic performance were collected; however, overall teacher ratings were used as the primary measure of academic performance and were the only measure reported in relation to mobility. Literacy, numeracy, and overall academic performance were rated by teachers as ‘low academic performance’ if students were ‘far below age level’ or ‘somewhat below age level’. Over half the Indigenous students aged 4–17 years (57.5%, 95%CI 54.7–60.3%) were rated by their teachers as having ‘low academic performance’. Indigenous students with high mobility were less likely to be rated as having ‘low academic performance’ compared to students who had lived in fewer homes after adjusting for family and household environment factors such as level of geographic isolation, child sex and age, and parenting quality (OR = 0.73, 95%CI 0.57–0.92). The authors note that this finding was unexpected.

Dockery et al. used three items from the PEDS to examine mobility and child learning and development [28]. Caregivers of children aged 0–7 years were asked: ‘Do you have any concerns about how (study child) understands what you say to her?’, ‘Do you have any concerns about how (study child) is learning to do things for herself?’, and ‘Do you have any concerns about study child’s learning or development?’. Response options were yes/a little/no. There was no difference in caregiver concerns about their child’s learning for families classified as having high mobility (one or more moves per year of child age) compared to low mobility families (β = 0.68, *p* = 0.210).

One study examined developmental risk by high mobility [23]. Chando et al. used caregiver PEDS data for Aboriginal children aged less than eight years to assess developmental risk. Parental concerns were grouped into 10 domains: global/cognitive; expressive language and articulation; receptive language; fine motor; gross motor; behavior; social and emotional; self-help; school; other. Developmental risk was categorized as low/no developmental risk = no predictive concerns, moderate = 1 predictive concern, and high = 2 or more predictive concerns [23]. Children with high mobility (four or more houses since birth) had higher odds of moderate developmental risk compared to children with low mobility (one house since birth) (OR = 1.81, 95%CI: 1.02–3.21). This association persisted after adjusting for the recruitment health service, child age and sex, and children (Adj.OR = 1.82, 95%CI 1.01–3.31). Similarly, children with high mobility had higher odds of high developmental risk compared to children with low mobility (OR = 5.19, 95%CI 2.78–9.68). In the full model (adjusting for the same factors above), the odds of high developmental risk remained fourfold higher for children with high mobility (Adj.OR = 4.41, 95%CI 2.25–8.63). Further analyses were conducted adjusting for broader factors associated with developmental risk, including the child’s sex, age, ear infections, whether the child was living out of home, and caregiver psychological distress. Again, children with high mobility had higher odds for moderate (Adj.OR = 1.75, 95%CI 0.95–3.23) and high developmental risk (Adj.OR = 4.13, 95%CI 2.04–8.35) compared to children with low mobility. Analyses indicated that odds of moderate or high developmental risk increased incrementally for each additional house lived in regardless of the child’s age.

#### 3.6.2. Qualitative Studies

One qualitative study reported on learning and development [24]. Participants from the Anderson et al. study described multi-family households struggling to cope with insufficient space, privacy, and basic amenities. Inadequate playing spaces were said to limit social and developmental opportunities. Frequent relocation, particularly due to homelessness, was also identified as unsettling for children and associated with further disruption to their schooling and learning [24].

### 3.7. Critical Appraisal

Six out of eight studies were rated as high quality, and two were rated as medium on the CASP (Table 3). All studies identified a range of socio-economic confounders in their analysis, had good sample sizes, and reported results that were applicable to local populations. Although study quality was rated high for the majority of quantitative studies, when interpreting the results, it is important to note a range of factors. Three studies reported study results in the text but did not provide detail on statistical analyses or confidence intervals [27,28,29]. Mobility was not the main exposure of interest in any of the six studies; rather, it was examined as a covariate. Other housing characteristics and environmental factors were examined, including housing tenure, house type, crowding, the condition of the house, the livability of the neighborhood, and the level of isolation. The broad range of socio-demographic and other confounders meant findings related to mobility and associated health outcomes were difficult to find and, in some instances, were combined with other housing-related findings. Results also need to be interpreted according to the accuracy of the housing mobility classification. Three of the six quantitative studies reported on the total number of house moves from 4–17 years [22,26,27]. Study findings, therefore, group together children who are different ages, for example, comparing the outcomes of a five-year-old child who has moved four times to the outcomes of a 16-year-old who has moved four times. Using moves per year of age would provide greater clarity on how high mobility impacts children at different developmental ages and stages and identify any longer-term implications. One of these studies noted these limitations [22].

### 3.8. Cultural Appraisal

Ethical codes recognize the importance of Indigenous research being community-driven and led [31]. International research guidelines emphasize the need for researchers to work with Indigenous communities at all stages of the research, to identify research questions, to work collaboratively in the design and conduct of the study, and to disseminate and translate findings into practice [21,32].

Half the studies in this review were responding to a need or priority determined by the community [22,23,24,25]. Six of the eight studies reported some level of Indigenous consultation, governance, or leadership, although the level of Indigenous input across each stage of the study was often unclear [22,23,24,25,26,27]. Few of the quantitative studies mentioned Indigenous research paradigms or working in collaboration with the community on the interpretation of findings. Many of the studies failed to employ approaches and/or assessments that were culturally developed, adapted, or tested for use with Indigenous children. One study used a culturally adapted version of the SDQ [26], and another reported findings using a strengths-based approach to report factors associated with good mental health in Indigenous children [22]. No studies reported on agreements to protect Indigenous intellectual or cultural property. Two of the studies did not report collaboration or input with Indigenous communities [28,29].

Two qualitative studies described culturally appropriate study designs, used Indigenous research paradigms, and involved the community in the interpretation of findings. Lowell et al. reported the most comprehensive Indigenous input [25]. The study was initiated in response to community concerns; video recordings of participants enabled a reflexive process for participants to explore footage, and interviews were conducted with community members to further explore findings. Yolŋu researchers from the study community were part of the research team, and their knowledge, perspectives, and interpretation of findings were described as valuable to the research process. They shared experiences and challenges as community members and local language and cultural knowledge. Their existing connections to Yolŋu in the community through kinship and other cultural systems enabled a level of engagement that outsiders could not achieve [25].

## 4. Discussion

This review identified eight studies that examined high mobility and associations with children’s health, developmental, and educational outcomes in Australian and Canadian Indigenous children. Three different measures of high mobility were used across the six quantitative studies. High mobility was defined as (1) one or more moves per year of child age [25,30], (2) living in five or more homes over their lifetime [31,33], and (3) living in four or more homes over their lifetime [23,29]. Three of these studies used a measure based on the number of homes children had lived in from birth to 17 years [22,26,27]. As Zubrick et al. noted, there is an age effect when measuring mobility in this way, as older children have a longer period in which to move. The wide age range of children meant that child health outcomes of older children (17 years) were being compared to child health outcomes of younger children (4 years), with the same number of house moves. William et al. also noted that having moved five times when you are 16 years old is not the same as having moved five times before you are six. The age at which children move house is important and may have different implications in terms of developmental ages and stages [34,35,36]. Although child health outcomes were reported for children between 4 and 17 years, the three studies also reported on child health outcomes by age bracket. Williamson et al. reported on social and emotional outcomes for 4–11 years [22], Zubrick et al. reported on social and emotional outcomes for 4–7 years and 8–11 years [26], and the second Zubrick et al. study [27] reported on learning and development outcomes for years 1–7 at school. An assessment of mobility as moves per year of age may have resulted in different findings. Further research is needed to fully understand the impact of residential moves for different developmental stages, particularly in longitudinal studies.

Inconsistency in the way housing mobility is measured and defined is reflected in the different proportions of Indigenous families with high mobility within each sample. Proportions of high mobility in each study ranged from 10% [29] to 37% [22]. Despite these differences, the reviewed studies reveal that a significant number of Indigenous children experience high housing mobility. Dockery et al. and Zubrick et al. report that the Indigenous children were more mobile than non-Indigenous children in their cohort. Lowell et al. and Anderson et al. both reported frequent relocation of families. One study has noted that families living in regional and urban areas had the greatest proportion of Indigenous families with high mobility compared to remote [24], and this has been supported by a later paper utilising SEARCH data [33]. This is supported by a growing body of research showing that Indigenous people are increasingly moving to urban areas and experiencing higher levels of mobility compared to families living in remote areas [8,37,38,39,40].

The review identified eight studies that examined four child health outcomes in relation to high mobility in Indigenous populations: physical health, social and emotional behavior, learning and development, and developmental risk. A review of study findings provides limited evidence for potential links between high mobility and children’s health, developmental, and educational outcomes. An association between high housing mobility and poorer emotional and behavioral outcomes was the most robust finding, supported by three studies. Three studies relied on caregiver SDQ [22,26,27], and one study used two items from the PEDS [28]. Two studies observed poorer emotional and behavioral outcomes for younger children within the cohort. Zubrick et al. reported an association for children aged 4–11 years but not for children aged 12–17 years [26]. Similarly, Williamson and colleagues reported an association for children aged 4–7 years but no difference for children aged 8–11 years. Due to the imprecise measurement of mobility, it is not clear whether older children’s outcomes were not impacted due to having a longer period between moves or if mobility has a greater impact on younger children’s development. However, findings are consistent with research suggesting that early childhood is a sensitive time period for high mobility, with challenges during this period more likely to impact on children’s development and well-being [5,9,15,16,36]. Results are also consistent with a 2021 meta-analysis that reported childhood mobility to be more strongly associated with social and emotional outcomes, such as externalizing and internalizing behaviors, than with physical health problems [12].

The association between high mobility and developmental outcomes was investigated by a single study in this review. Mobility was just one variable amongst a range of factors related to increased developmental risk, such as ear health, caregiver mental health, and being in out-of-home care. The study clearly demonstrated a linear relationship between the number of houses the child lived in since birth and developmental risk. The PEDS measure used to assess developmental risk covered 10 domains inclusive of physical health, social/emotional, learning, and development outcomes. The findings suggest that significant numbers of Indigenous Australian children could be at increased developmental risk due to high mobility. This highlights an urgency to better understand the impacts on Indigenous children, the reasons underpinning higher mobility, and what families want/need to support their children in the context of housing instability.

Investigation of high mobility and physical health was insufficient and inconclusive [28,29]. There was evidence of poorer global health in a study using an SF-36 global health item [29]; however, a second study using the same measure found no difference [28]. One study reported that high mobility was associated with ear issues, but this was not significant after adjusting for broader housing issues than mobility [29].

There was no clear evidence of associations between high mobility and learning and development, including school attendance and academic performance. One study examined school attendance and academic performance [27], and a second study used the caregivers’ concerns for their child’s learning and development (PEDS) [28]. Measures and approaches that are not tailored for Indigenous children can have implications for interpreting the findings. For example, asking teachers to rate children’s academic performance by category relies on the perceptions of the teacher and does not consider differences in Indigenous children’s knowledge, cultural strengths, or experiences [41]. It should also be noted that there have been mixed reviews on the validity of using the standard SDQ with Indigenous children, particularly the prosocial behavior, peer problems, and conduct problem scales [42,43,44].

A key theme across all the studies was the multiple and interrelated aspects of housing disadvantage being experienced by Indigenous families. A wide range of housing challenges was identified, including insufficient housing availability, overcrowding, poor housing conditions, a lack of affordability, long waiting lists for government housing, and discrimination in the private rental market. Consideration of the multiple housing factors provides valuable insight into the lived realities of Indigenous families’ housing experiences, for example living with family and friends (including kinship groups) for extended periods of time or frequently moving from family to family or to transient or emergency accommodation. Direct health consequences of living in crowded houses include having limited access to food and house utilities and a lack of control of the home environment (e.g., the condition of the home, space for children, and privacy). There is a growing body of research showing that some low-income families are being forced to move. These moves are for housing-related reasons, such as leaving crowded, unaffordable, or poor-quality dwellings and being evicted or asked to leave by the landlord [4,45,46,47,48].

Findings are consistent with literature showing that Indigenous families face multiple forms of social and financial disadvantage [46]. Housing is a basic need and has been identified as an important social determinant of health [49]. The clustering of housing factors will generally coexist for families and have a cumulative impact on health and well-being, particularly with prolonged experience in childhood [47].

### Strengths and Limitations

This systematic review is the first to explore associations between residential mobility and Indigenous children’s health, development, and educational outcomes from 0 to 12 years. The study was conducted in a research team comprised of Aboriginal (2) and non-Aboriginal researchers (2) with experience in collaborative Aboriginal and Torres Strait Islander health research in partnership with Aboriginal and Torres Strait Islander communities in South Australia. The review was conducted by an Aboriginal doctoral candidate and will be used to inform their strengths-based analyses of data collected in the Aboriginal Families Study [50]. The study has been auspiced by the Aboriginal Health Council of South Australia (the peak body for Aboriginal community-controlled health organizations in South Australia) since 2007 and is overseen by a long-standing Aboriginal Advisory Group. This review examines Indigenous child health outcomes with the aim of recognizing and supporting healthy and strong Indigenous families, communities, and culture into future generations [51].

Several limitations were identified. The small number of studies identified, plus the lack of diverse data, limited the findings and evidence available for review. Two groups of studies shared the same administrative dataset—the WAACHS data [26,27] and the SEARCH cohort data [22,23]. The majority of studies identified were Australian-based (7/8), with no studies involving Indigenous New Zealand children identified.

All quantitative studies used measures of child outcomes based on caregiver reports or teacher assessments of child academic outcomes, with the majority relying on one or two items per domain. The use of direct assessment or multi-item standardized measures are the strongest indicators of child health and development. Using measures that are not culturally tailored for Indigenous children has significant implications for how children may perform/score. Further research is needed to create culturally appropriate standardized measures suitable for Indigenous children.

Positive child health outcomes associated with mobility were not examined. Cultural factors such as staying connected to family, kin, and country are important drivers of mobility in Indigenous communities [39,40,52]. Some families may be moving to be closer to extended family, to be on country, or to access cultural activities or community to better support the development and well-being of their children/family. (To be on country refers to an area to which people have a traditional or spiritual association, a sense of connection or belonging) [53]. The positives of different family structures and housing are also not addressed. The Lowell study described participants valuing support and connection with extended family, “everyone’s there for the little one”, while participants in the Anderson study reported living with extended family to avoid homelessness. Cultural and community factors are associated with health benefits for many Indigenous people [53]. This is centered on a broader holistic concept of health, including social, emotional, physical, cultural, and spiritual dimensions of well-being. Health is conceptualized as a balance between all these dimensions, and with extensions beyond the individual person to also reflect family and community wellness [31,53]. There are a number of tools now available to capture cultural determinants and measure their effect on health [54,55]. It is vital that future research explores why families are moving and considers broader cultural determinants of health that include potential benefits to child development and well-being [56].

## 5. Conclusions

This systematic review highlights the lack of research into the relationship between residential mobility and Indigenous children’s health, developmental and educational outcomes. The research suggests that early childhood is a sensitive time where challenges such as high mobility are more likely to impact children’s development and well-being. Further research is needed to fully understand the impact of residential moves for different developmental stages. More nuanced research that includes the drivers of high mobility and the identification of individual, family, school, community, and cultural factors that support positive outcomes for Indigenous children can guide more effective health, school, and policy responses. Prioritizing the involvement, collaboration, and empowerment of Indigenous communities and leadership is critical for future research [31].

## Figures and Tables

**Figure 1 ijerph-20-04332-f001:**
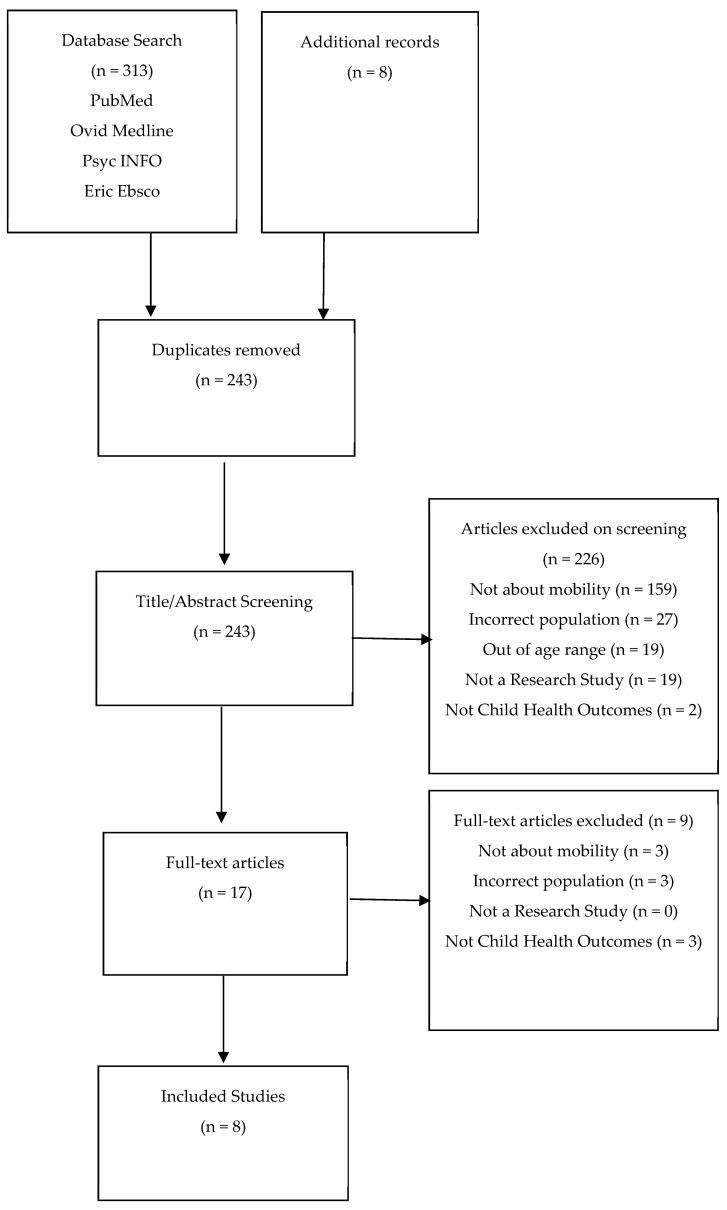
PRISMA Flow Diagram of study selection.

**Table 1 ijerph-20-04332-t001:** Characteristics of the included studies (n = 8).

First Author	Study Type	Study	Country	Year	Setting	Sample	Indigenous	Child Age (Years)	Data Collection Method
Quantitative									
Zubrick [26]	Cross-sectional	WAACHS _1_	Australia	2005	Community	3993	3993	4–17	Parent/Caregiver Survey
Zubrick [27]	Cross-sectional	WAACHS _1_	Australia	2006	Community	2379	2379	4–17	Parent/Caregiver Survey Teacher/School Principal Survey Child Assessment administered by teacher
Dockery [28]	Prospective cohort study	LSIC _2_	Australia	2013	Community	B-cohort: 960 K-cohort: 727	960 727	0–7	Parent/Caregiver Survey
Williamson [22]	Cross-sectional	SEARCH _3_	Australia	2016	Community	1005	1005	4–17	Parent/Caregiver Survey
Guévremont [29]	Cross-sectional	ACS _4_	Canada	2016	Community	3640	3640	2–5	Parent/Caregiver Survey
Chando [23]	Cross-sectional	SEARCH _3_	Australia	2020	Community	725	725	0–8	Parent/Caregiver Survey
Qualitative									
Andersen [24]	Qualitative		Australia	2016	Community	38 adults	35 adults	0–18	Focus Groups
Lowell [25]	Qualitative		Australia	2018	Remote Community	6 families 30 community adults	6 children	0–7	Parent/Caregiver Interviews Family Member Interview Video Recording

_1_ West Australian Aboriginal Child Health Survey; _2_ Longitudinal Study of Indigenous Children; _3_ Study of Environment on Aboriginal Resilience and Child Health; _4_ Aboriginal Child Survey.

**Table 2 ijerph-20-04332-t002:** (**a**) Definition of mobility and child health outcomes reported in each study (Quantitative). (**b**) Definition of mobility and child health outcomes reported in each study (Qualitative).

(**a**)							
**First Author**	**Child (Years)**	**Definition of High Mobility**	**High** **Mobility = n (%)**	**Physical Health**	**Social Emotional**	**Learning Development**	**Measures**
Zubrick (2005) [26]	4–17	≥5 moves over child’s lifetime	3993 (27.4%)		x		Strengths and Difficulties Questionnaire (Modified) -Parent/caregiver report
Zubrick (2006) [27]	4–17	≥5 moves over child’s lifetime	Not reported			x	Principal Report: ≥26 absences per year ≥10 unexplained absences per year Teacher classification of academic performance
Dockery [28]	0–7	≥1 move per year of life	1687 (20%)	x	x	x	General Health Status item (SF-36) Parent Evaluation of Development: 2 Items Social emotional development 3 Items Learning/development
Williamson [22]	4–17	≥4 homes previously lived in over child’s lifetime	327 (37%)		x		Strengths and Difficulties Questionnaire -Parent/caregiver report
Guévremont [29]	2–5	≥1 move per year of life	364 (10%)	x	x		Strengths and Difficulties Questionnaire -Parent/caregiver report General Health Status item (SF-36) Single Item Measures: Ear Infections per year Chronic conditions Activity Limitation (yes/no) Chronic Respiratory Conditions (yes/no) Serious Injury in the past year (yes/no)
Chando [23]	0–8	Number homes lived in (1, 2, 3, 4+) (Number with 4+ moves reported)	111 (15%)			x	Parent Evaluation of Development Status 10 Items
(**b**)							
**First Author**	**Child Age (Years)**	**Respondent**	**Housing Mobility Definition**	**Physical Health**	**Social Emotional**	**Learning Development**	**Concerns Raised for Children Relating to Housing Mobility**
Anderson [24]	0–18	Mothers, Fathers, Grandmothers, Health service staff and clients	Secondary homelessness: frequent relocation, house hopping, and precarious housing.	x	x	x	Concerns with social and emotional behavior Otitis Media or ear infections Asthma and respiratory conditions Injury Risk Disrupted schooling Inadequate sleep High rates of communicable disease Chronic stress of parent/caregiver
Lowell [25]	0–7	Family and community members	Housing insecurity and moving from house to house: insecure and insufficient housing.	x	x	x	Physical health Behavioral concerns Children’s safety and the potential for accidents Disrupted and inadequate sleep Sharing sickness Limited access to kitchen, bathroom, laundry facilities Food security Conflict/stress amongst families and children

**Table 3 ijerph-20-04332-t003:** Critical and cultural appraisal of the included studies (n = 8).

First Author	Critical Appraisal Quality	Cultural Appraisal Quality
Quantitative		
Zubrick 2005 [26]	Medium	Medium
Zubrick 2006 [27]	Medium	Medium
Dockery [28]	High	Low
Williamson [22]	High	High
Guévremont [29]	High	Low
Chando [23]	High	High
Qualitative		
Andersen [24]	High	High
Lowell [25]	High	High

## Data Availability

Not applicable.

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
