# Peer review of "A Systematic Review of Child Health, Developmental and Educational Outcomes Associated with High Mobility in Indigenous Children from Australia, Canada and New Zealand"

_ijerph, 2023, doi:10.3390/ijerph20054332_

Round 1

Reviewer 1 Report

I found your article to be highly informative, thoughtful and extremely well written. The research is thorough and clearly explains the considerations of the Indigenous context of your own work and the studies you analyzed. The information in your article is concise and clear throughout, maintaining high readability and interest as you set out highly detailed discussion.

1.  Do you consider the topic original or relevant in the field?  Does it address a specific gap in the field?

Yes, this is an orginal and highly relevant topic, and addresses a specific gap in the field.  It's consideration of the effects of high mobility on Indigenous child health and educational outcomes is one that has not had a great deal of research done, and is of great importance with real-life consequences for Indigneous peoples.  The paper addresses the Indigenous peoples in Australia, New Zealand and Canada for an effective comparative approach, while providing a very clear justification for the comparison of these three jurisdictions.  It provides an analytical discussion on studies and articles on these issues, pointing out their strengths and weaknesses, drawing out relevant conclusions which can be drawn and highlighting where changes in research approaches are needed to make the results more meaningful for and respectful of Indigenous communites, and where there are gaps in the existing research.  The systematic review provides a consideration of both qualitative and quantitative studies.  The inclusion of both kinds of methodologies within the systematic review, as well as the inclusion of studies on Canada, New Zealand and Australia provide a comprehensive and in depth critical consideration of the work done on the issue of mobility, educational achievements and Indigenous child health.  While it provides a broad view, the systematic review also provides one that is in depth, and thus a rich and unique analysis of this topic.

2.  What does it add to the subject area compared with other published material?

It points out the importance of an Indigenous context and understanding of studies and research done, and highlights how little this has been included within other studies done.  The shortcomings of other studies are discussed within a framework of explaining why further understanding is needed and what impacts this can have on Indigenous children and Indigenous communities.  Clear definitions and data sets are included, with a sensitivity to Indigenous context.  The systematic review provides both breadth and depth in the way that a single study does not, and the inclusion of studies from these 3 states, which themselves share many commonalities about their Indigenous populations, provides a focus that other studies have not.  This provides a richly informed discussion and a great deal of insight into this topic.

3.   What specific improvements should the authors consider regarding the methodology?   What further controls should be considered?

No suggestions for any improvements on the methodology .

4.    Are the references appropriate?

Yes, it is thoroughly referenced, and the references are relevant and appropriate.

5.    Please include any additional comments on the tables and figures.

The tables and figures are well set out, clear, and complement the written material.    They enhance the written information and help to provide clear information.

Reviewer 2 Report

This review covers a very important topic, and I am grateful to the authors for their excellent work. My comments are brief and provided below, with reference to the relevant line numbers.

Line 98: Can an example of a strategy to better support families and children affected by high mobility be given?

Line 142-143: Consider mentioning the differences between the three Indigenous populations.

Line 241-242: you mention use of the SDQ. Can you make a comment on its validity for use with the Indigenous Australian population? I note that you do make reference to cultural appropriateness at lines 523, 665-666. This perhaps should be mentioned earlier.

Line 353: Section 3.4.2 (Qualitative studies) was interesting. However, it was about poor housing conditions, which is distinct from high mobility (the topic of this systematic review), although related. Can you relate the topic of poor housing in this section to high mobility. You do mention this at lines 630-632. This could be mentioned in section 3.4.2.

Line 617-619: good to see acknowledgement of cultural differences when assessing academic performance of Indigenous kids. (no action required by authors).

Line 672: Do readers not familiar with Indigenous Australians know what ‘to be on country’ means? Maybe elaborate briefly?

Minor changes

Line 65: needs a comma after ‘However’

Line 439: comma needed after ‘however’

Line 614-615: apostrophe after ‘caregivers’

Reviewer 3 Report

A good solid systematic review - perhaps a little limited in scope making the results and conclusions less impactful. Definitely a need for more research looking at this topic  - esp. in Indigenous populations and children.

All other comments and minor edits in the attached pdf.
